# TRANSFER ALIGNMENT NETWORK FOR DOUBLE BLIND UNSUPERVISED DOMAIN ADAPTATION

## ABSTRACT

How can we transfer knowledge from a source domain to a target domain when each side cannot observe the data in the other side? The recent state-of-the-art deep architectures show significant performance in classification tasks which highly depend on a large number of training data. In order to resolve the dearth of abundant target labeled data, transfer learning and unsupervised learning leverage data from different sources and unlabeled data as training data, respectively. However, in some practical settings, transferring source data to target domain is restricted due to a privacy policy.

In this paper, we define the problem of unsupervised domain adaptation under *double blind* constraint, where either the source or the target domain cannot observe the data in the other domain, but data from both domains are used for training. We propose TAN (Transfer Alignment Network for Double Blind Domain Adaptation), an effective method for the problem by aligning source and target domain features. TAN maps the target feature into source feature space so that the classifier learned from the labeled data in the source domain is readily used in the target domain. Extensive experiments show that TAN 1) provides the state-of-the-art accuracy for double blind domain adaptation, and 2) outperforms baselines regardless of the proportion of target domain data in the training data.

## 1 INTRODUCTION

How can we minimize data distribution gap between source and target domains under constraint of data visibility, such that each side cannot observe the data in the other side? Recent state-of-the-art deep learning algorithms perform well on test data when given abundant labeled training data. However, in real applications, collecting a large amount of labeled data needs tedious labor work while unlabeled data are abundant. Given a certain number of unlabeled data from a target domain, unsupervised transfer learning exploits the knowledge from a related source domain, which has enough unlabeled data accompanying small number of labeled data, to improve the model performance on target domain. Since the source domain is related to but different from the target domain, data distribution shift exists between the two domains. Domain adaptation attempts to reduce the data distribution shift by effectively transferring knowledge from the source to the target domains. Recent state-of-the-art methods focus on digging out domain-invariant features, and learn more transferable representations to improve transferability. Cao et al. (2018) proposed transfer learner by extracting invariant feature representations while Long et al. (2018) applied generative adversarial model and generated feature extractor to confuse domain classifier. The premise of applying such methods is that source and target data are visible simultaneously. However, in real-world settings including hospital cases, it is not allowed to access source and target data simultaneously for privacy reason.

In this paper, we define the problem of unsupervised domain adaptation under *double blind* constraint which restricts mutual accesses between source and target domains. As shown in Figure 1, general domain adaptation can leverage source and target data simultaneously, but domain adaptation under double blind constraint cannot utilize the two data simultaneously because of the restriction on data accessibility. Due to the constraint, existing methods on domain adaptation are not suitable for the new problem. We are exposed to two challenges: (1) since source and target data are not visible simultaneously, we have no way of knowing how much the data distribution shifts between the two domains, and (2) the target data cannot be used to train a meaningful classifier due to the missing label.

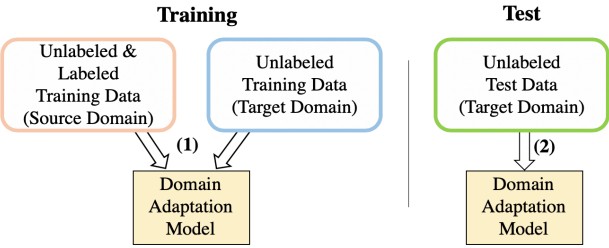

(a) Domain adaptation without double blind constraint

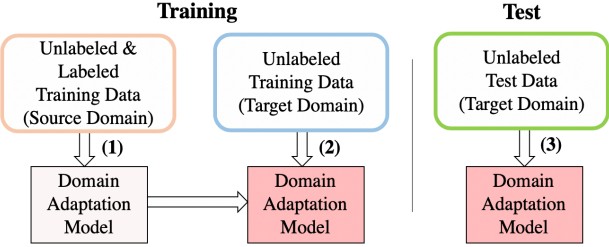

(b) Domain adaptation under double blind constraint

Figure 1: Comparison between domain adaptation with or without double blind constraint. (a) Domain adaptation without double blind constraint leverages source and target data simultaneously at training time. (b) Domain adaptation with double blind constraint cannot feed data from source and target domains simultaneously into model at training time.

To address the challenges, we propose TAN (Transfer Alignment Network for Double Blind Domain Adaptation), an effective method for double blind domain adaptation which resolves the data distribution shift problem by aligning source and target feature spaces. TAN maps target features into source feature space in order to utilize the classifier trained in source domain to target data. The experimental results show the superior performance of TAN.

Our contributions in this paper are the followings:

- **Problem Definition**. We propose a new problem of unsupervised domain adaptation under double blind constraint, which involves data inaccessibility during training process.
- **Method**. We introduce a simple but effective model TAN which aligns data distribution of source and target domains.
- **Experiments**. Extensive experiments on real-world multivariate datasets show that TAN provides the state-of-the-art performance.

In the rest of this paper, we describe preliminaries, proposed method, experimental results, and related works, and conclude in the end.

## 2 PRELIMINARIES

In this section, we provide brief description of autoencoder pre-training, and transfer learning with fine-tuning.

### 2.1 AUTOENCODER PRE-TRAINING

In order to leverage unlabeled data in neural network training, the most common approach is to initialize parameters by weights pre-trained using autoencoder (Erhan et al. (2010)). Autoencoder (Bengio et al. (2009)) is trained to reconstruct feature vector from code constrained either in dimensionality or sparsity. After training, the encoded vector from each input feature has enough information to reconstruct the original feature, and represents the hidden structure in the input feature space.

Autoencoder is composed of two parts: encoder and decoder. Encoder maps the input feature into a constrained code, and decoder maps the code back to feature reconstruction. Given input feature vector $x$, encoder $f$ maps $x$ to code $h = f(x)$. The decoder $g$ maps $h$ to reconstruction $\hat{x} = g(h) = g(f(x))$. Both encoder and decoder are defined as neural networks, with iterative applications of affine transformation and activation function. Then the model is trained to minimize the reconstruction error for each data point $d_i$ in dataset. The objective function is defined as $\min_\theta \sum_i ||\hat{x}_i - x_i||^2$ where $\hat{x}_i$ is a reconstruction $g(f(x_i))$ from input $x_i$ and $\theta$ denotes neural network parameters for encoder $f$ and decoder $g$.

Given large amount of unlabeled data and few labeled data, we use code vector $f(x)$ from encoder as input in a classifier to effectively train it using the few labeled data. If the code vector has enough information to reconstruct the input feature vector, then it would contain enough information for the classifier as well.

## 2.2 TRANSFER LEARNING WITH FINE-TUNING

Transfer learning with fine-tuning approach is simple and easy to apply in neural network training. In the training process, a neural network classifier learns useful intermediate hidden units (Bengio (2012)). These hidden units provide enough information for simple classifier to achieve the state-of-the-art performance not only in the same task (Sharif Razavian et al. (2014)) but also in related tasks (Sermanet et al. (2013)). This property has been exploited to use neural network trained on a source domain as initialization for further fine-tuning on a target domain. With careful analysis (Yosinski et al. (2014)), it has been shown that transferring pre-trained neural network features improves performance. Fine-tuning is important to transform source-specific features into target-specific features and to alleviate layer co-adaptation problem. It has been a common practice in neural network transfer learning to adapt parameters that are pre-trained from a source domain and fine-tune the model on a target domain.

## 3 PROPOSED METHOD

In this section, we precisely define the problem of double blind unsupervised domain adaptation and its challenges. Then we describe our method TAN in detail. Figure 2 illustrates the architecture of our method in detail and Table 1 provides the definitions of symbols used.

## 3.1 PROBLEM DEFINITION AND CHALLENGES

In unsupervised domain adaptation, a source domain has abundant unlabeled data $X^{s,u} = \{x_i^{s,u}\}_{i=1}^{n_{s,u}}$ and small number of labeled data $X^{s,l} = \{(x_i^{s,l}, y_i^{s,l})\}_{i=1}^{n_{s,l}}$. A target domain has only limited number of unlabeled data $X^{t,u} = \{(x_i^{t,u})\}_{i=1}^{n_t}$. Note that the number $n_{s,u}$ of unlabeled samples from the source domain is larger than the number $n_t$ from the target domain. Both of source and target data contain the same number K of classes. The source and the target data are sampled from distributions $p$ and $q$, respectively, where $p \neq q$. Our goal is to learn a classifier which performs well in target classification task under *double blind* constraint, where each domain cannot observe the data in the other domain, by leveraging both data in the source and target domains. Note that this problem of *double blind* domain adaptation is slightly different from *blind* domain adaptation (Uzair & Mian (2017)) since the former uses the data from both source and target domains for training, while the latter uses the data only from the source domain for training.

Due to the double blind constraint, 1) existing domain adaptation models cannot be implemented directly, and 2) the distribution discrepancy between source and target data is not measured easily. Another complication is that the target data have no labels, which makes it impossible to train a classifier if only using the target data without accessing the labeled data at the source domain. We need a novel method to match feature distributions of source and target domains.

## 3.2 TRANSFER ALIGNMENT NETWORK FOR DOUBLE BLIND DOMAIN ADAPTATION

The target data cannot be used to generate a classifier due to missing labels, which forces us to utilize the source domain data to train a classifier and make use of it in the target domain. Even

Table 1: Table of symbols.

| Symbol | Terminology | Description |
|---|---|---|
| $X^{s,u}$ | source unlabeled data | data instances without labels in source domain |
| $X^{s,l}$ | source labeled data | data instances with labels in source domain |
| $X^{t,u}$ | target unlabeled data | data instances without labels in target domain |
| $f_{encode}$ | encoder | neural network that encodes input into code vector |
| $f_{decode}$ | decoder | neural network that decodes code vector into input reconstruction |
| $f_{align}$ | transfer aligner | neural network that aligns target code vector to source code vector |
| $f_{classify}$ | classifier | neural network that classifies code vector |
| $f_{s,encode}$ | source encoder | encoder after the Step 2 training |
| $f_{s,classify}$ | source classifier | classifier after the Step 2 training |
| $f_{t,encode}$ | target encoder | encoder after the Step 3 training |
| $f_{s,encode}(X)$ | source code space | space of code from source encoder |
| $f_{t,encode}(X)$ | target code space | space of code from target encoder |

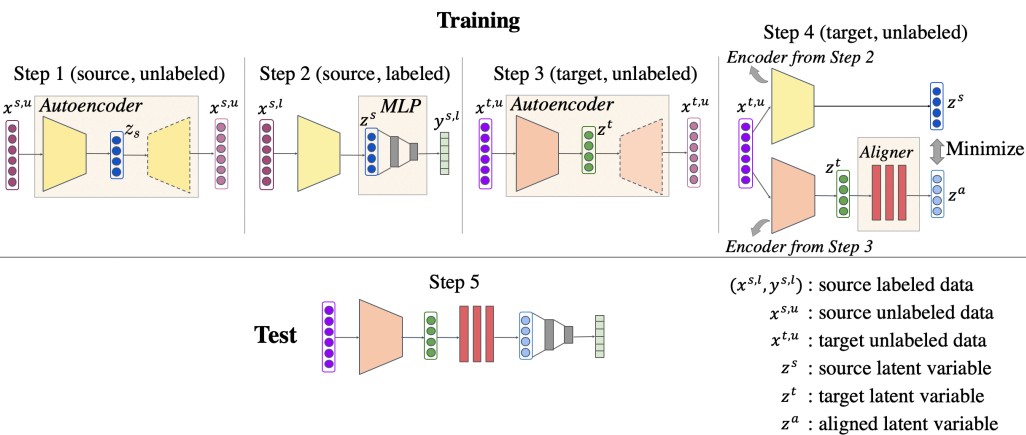

Figure 2: Structure of TAN. In training step 1, we train an autoencoder with unlabeled source instances. In training step 2, we train a stack of classifier and the encoder using labeled source instances. Then, after transferring the model to a target domain, we fine-tune the autoencoder on unlabeled target instances in training step 3. In training step 4, a transfer aligner (in red color) is trained to map the target feature space back to the source feature space using unlabeled target instances. At test time, each instance goes through the final model consisting of the fine-tuned autoencoder and the transfer aligner trained at step 4, and the source classifier trained at step 2.

though there are a large amount of data in the source domain, most of them are unlabeled; it is hard to train a good classifier using only the small amount of labeled data in the source domain. Thus, we use the source unlabeled data to train an autoencoder to extract meaningful features. Then, the source labeled data go through the autoencoder to become valuable features, and we utilize the features with labels to train a classifier. Due to the double blind constraint, we can only transfer the model trained in the source domain, not data. After the model is transferred to the target domain, we finetune the autoencoder with the target unlabeled data. Because of data distribution discrepancy between the source and the target data, directly feeding the extracted target domain features into the pretrained classifier will seriously damage the accuracy. To tackle this problem, we add and train transfer adaptation layers, which we call "transfer aligner", between the autoencoder and the classifier using only the target unlabeled data. After being passed through the transfer aligner, the target domain features have a distribution as close to the source distribution, so that we can use the classifier trained in the source domain to classify the target unlabeled data.

---

**Algorithm 1** TAN (Transfer Alignment Network for Double Blind Domain Adaptation)

---

**Input:** source labeled data $X^{s,l}$, source unlabeled data $X^{s,u}$, and target unlabeled data $X^{t,u}$
**Output:** encoder for autoencoder $f_{encode}$, transfer aligner $f_{align}$, and classifier $f_{classify}$
 1: (Step 1) Train $f_{encode}$ and $f_{decode}$ on $X^{s,u}$ with L2 reconstruction loss
 2: (Step 2) Train $f_{encode}$ and $f_{classify}$ on $X^{s,l}$ with cross-entropy classification loss
 3:         Take snapshot $f_{encode}$ as $f_{s,encode}$
 4: (Step 3) Train $f_{encode}$ and $f_{decode}$ on $X^{t,u}$ with L2 reconstruction loss
 5:         Take snapshot $f_{encode}$ as $f_{t,encode}$
 6: (Step 4) $f_{align} = TrainAligner(f_{s,encode}, f_{t,encode}, X^{t,u})$ in Algorithm 2

---

**Algorithm 2** Train Transfer Aligner

---

**Input:** source encoder $f_{s,encode}$, target encoder $f_{t,encode}$, and target unlabeled data $X^{t,u}$
**Output:** transfer aligner $f_{align}$
 1: Initialize $f_{align}$ with random weights
 2: Calculate source code $z_i^s = f_{s,encode}(x_i)$ and target code $z_i^t = f_{t,encode}(x_i)$ for each data instance $x_i \in X^{t,u}$
 3: Train $f_{align}$ to minimize $\sum_{i=1}^{n_t} ||z_i^s - z_i^a||^2$ using gradient based optimization, where $z_i^a = f_{align}(z_i^t)$

---

The whole architecture is simple but effective, consisting of encoder, transfer aligner, and multilayer perceptron. Figure 2 shows the training and the test steps of TAN. We train an autoencoder with unlabeled source instances in Step 1. The blue rectangle is a feature vector extracted from the encoder. In Step 2, we utilize the encoder part trained from Step 1 by adding multilayer perceptron on top of the code vector and the model is trained with labeled source instances. In Step 3, we transfer the model to the target domain, but only fine-tune the autoencoder using unlabeled target instances. In Step 4, the transfer aligner (in red) is trained to directly map the target code vector into source code space using the unlabeled target instances. We train the transfer aligner $f_{align}$ to map the code of the target encoder $f_{t,encode}$ to the code of the source encoder $f_{s,encode}$ for the target unlabeled data $X^{t,u}$. As shown in Figure 2, the encoder, the transfer aligner, and the classifier are stacked together for the complete model. We test the final model with unlabeled target test data. The whole algorithms are described in Algorithms 1 and 2. We further elaborate each step as follows:

- **Step 1.** The encoder $f_{encode}$ and decoder $f_{decode}$ are trained on source unlabeled data $X^{s,u}$. The model is trained to minimize the following autoencoder reconstruction objective function.

$$\sum_{x_i \in X^{s,u}} ||x_i - f_{decode}(f_{encode}(x_i))||^2 \tag{1}$$

- **Step 2.** We train a classifier $f_{classify}$ on top of the previous encoder $f_{encode}$ on source labeled data $X^{s,l}$ with fine-tuning. The training process minimizes the following cross-entropy objective function.

$$- \sum_{x_i \in X^{s,l}} \sum_{k \in K} y_k \log f_{classify}(f_{encode}(x_i))_k \tag{2}$$

  We call the encoder $f_{encode}$ and the classifier $f_{classify}$ at this step as source encoder $f_{s,encode}$ and source classifier $f_{s,classify}$, respectively. Snapshot of current parameters of the source encoder $f_{s,encode}$ is stored separately.
- **Step 3.** The encoder $f_{encode}$ and the decoder $f_{decode}$ is fine-tuned on target unlabeled data $X^{t,u}$ again with the following standard autoencoder reconstruction objective function.

$$\sum_{x_i \in X^{t,u}} ||x_i - f_{decode}(f_{encode}(x_i))||^2 \tag{3}$$

  We call the encoder $f_{encode}$ with fine-tuned parameters at this step as target encoder $f_{t,encode}$. Snapshot of current parameters of target encoder $f_{t,encode}$ is again stored separately.

Table 2: **Statistic of the datasets.**

|  | HIGGS[1] | HEPMASS[2] | SUSY[3] | Sensorless[4] | Gas[5] |
|---|---|---|---|---|---|
| Number of instances | 11,000,000 | 10,500,000 | 5,000,000 | 58,509 | 13,910 |
| Feature dimensions | 28 | 28 | 18 | 49 | 128 |
| Number of classes | 2 | 2 | 2 | 11 | 6 |

- **Step 4.** The transfer aligner $f_{align}$ is trained to map target code $f_{t,encode}(x)$ to source code $f_{s,encode}(x)$ on the target unlabeled data $X^{t,u}$. The objective function is L2 distance between the source code $f_{s,encode}(x_i)$ and the mapped target code $f_{align}(f_{t,encode}(x_i))$:

$$\sum_{x_i \in X^{t,u}} ||f_{s,encode}(x_i) - f_{align}(f_{t,encode}(x_i))||^2 \tag{4}$$

In this step, only the transfer aligner $f_{align}$ (the red part) is trained, while the encoders are fixed.

# 4 EXPERIMENTS

We present experimental results to answer the following questions.

- **Q1. (Predictive Performance)** What are the accuracies of TAN and competitors for double blind domain adaptation?
- **Q2. (Effect of Transfer Aligner Training)** Does the benefit of transfer aligner come from alignment of features rather than deployment of more layers?
- **Q3. (Effect of Amount of Target Data)** Is the model affected by the proportion of source and target data?

## 4.1 EXPERIMENTAL SETTING

**Datasets.** We conduct experiments on five multivariate datasets summarized in Table 2: HIGGS[1], HEPMASS[2], SUSY[3], Sensorless[4], and Gas[5]. HEPMASS, HIGGS and SUSY are sensor observations of particle collision experiments where labels are binary and denote the actual generation of the particles of interest. Sensorless and Gas are also sensor observations where labels denote the diagnosis of motor status and the type of gas in chemical reaction, respectively.

**Data preprocessing.** In order to perform the domain adaptation experiments, we require a dataset with source/target labeled/unlabeled instances. We divide the total training data into 10 clusters using K-means so that instances in each cluster are closer to each other compared to those in other clusters. We first select a random point $x_r$. Then, we set the farthest point $c_1$ from $x_r$ as the first centroid. The 2nd centroid $c_2$ is determined to be the point with the farthest distance to $c_1$. The $k$th centroid $c_k$ for $k = 3, 4, ..., 10$ is determined as the point with the farthest shortest distance to the existing centroids: $c_k = \arg\max_{x_i} \min(||x_i - c_1||_2, ..., ||x_i - c_{k-1}||_2)$. The found 10 clusters can be divided into source and target data with 9:1 ratio, which leads to 10 possible combinations of source and target data. We select the cluster with the largest H-divergence to other clusters as the target data. For efficient computation of H-divergence, we approximate it using Proxy A-distance $\hat{d}_A = 2(1 - 2\epsilon)$ where $\epsilon$ is the error of correctly discriminating source and target domains (Ben-David et al. (2010)) using SVM. Because target data instances are sampled from only one cluster, source and target datasets have different feature distribution as intended. The final data ratio is Source Unlabeled:Source Labeled:Source Test:Target Unlabeled:Target test = 80.1:8.9:1:9:1.

**Model settings.** We examine the influence of transfer aligner for domain adaptation with the following architecture: 80-40-20-10 autoencoder, 30-80-150 transfer aligner, and 300-300-300-300-

---

[1] http://archive.ics.uci.edu/ml/datasets/HIGGS
[2] http://archive.ics.uci.edu/ml/datasets/HEPMASS
[3] http://archive.ics.uci.edu/ml/datasets/SUSY
[4] http://archive.ics.uci.edu/ml/datasets/Dataset+for+Sensorless+Drive+Diagnosis
[5] http://archive.ics.uci.edu/ml/datasets/Gas+Sensor+Array+Drift+Dataset

Table 3: Test accuracy of TAN and baseline models (%). TAN gives the best accuracy.

| Methods | HIGGS | HEPMASS | SUSY | Sensorless | Gas |
|---------|-------|---------|------|-----------|-----|
| S(UL) | $83.74 \pm 0.62$ | $73.05 \pm 0.77$ | $84.99 \pm 0.83$ | $75.38 \pm 0.99$ | $70.53 \pm 0.54$ |
| S(UL)-T(U) | $59.97 \pm 0.22$ | $76.18 \pm 0.68$ | $48.18 \pm 0.63$ | $76.00 \pm 1.26$ | $68.28 \pm 1.08$ |
| TAN | $\mathbf{92.74 \pm 0.37}$ | $\mathbf{79.95 \pm 0.84}$ | $\mathbf{88.86 \pm 0.26}$ | $\mathbf{78.02 \pm 1.37}$ | $\mathbf{74.63 \pm 1.49}$ |

Table 4: Effect of transfer aligner in test accuracy (%). The superior accuracy of TAN comes from its transfer aligner, not from increased number of parameters.

| Methods | HIGGS | HEPMASS | SUSY | Sensorless | Gas |
|---------|-------|---------|------|-----------|-----|
| S(UL)-T(U)-Large | $87.05 \pm 0.39$ | $76.23 \pm 1.23$ | $84.42 \pm 0.42$ | $77.12 \pm 1.89$ | $72.29 \pm 1.34$ |
| TAN | $\mathbf{92.74 \pm 0.37}$ | $\mathbf{79.95 \pm 0.84}$ | $\mathbf{88.86 \pm 0.26}$ | $\mathbf{78.02 \pm 1.37}$ | $\mathbf{74.63 \pm 1.49}$ |

300 classifier. We apply ReLU activation function for all hidden layers(including hidden layers in encoder, decoder, transfer aligner, and mlp) and softmax function for the prediction layer while encoding layer ($z^s$ and $z^t$ in Figure 2) and reconstruction layer ($x^{s,u}$ and $x^{t,u}$ in Figure 2) have no activation function. We implement all methods based on the Tensorflow framework. All parameters are randomly initialized in the source domain, and the model is trained from scratch while we utilize the pretrained model from the source domain to fine-tune it in the target domain.

**Competitors.** We compare TAN with the following baselines.

- S(UL): a stack of an encoder and a neural network classifier trained using source data (unlabeled data and labeled data, respectively) and tested on target data without finetuning. This corresponds to using only Steps 1 and 2 of Figure 2 in training.
- S(UL)-T(U): this model retrains the S(UL) model with the target unlabeled data. This corresponds to using Steps 1, 2, and 3 of Figure 2 in training. In another point of view, this model is similar to TAN, but without the transfer aligner in Step 4.
- S(UL)-T(U)-Large: this model is similar to S(UL)-T(U), but contains more layers and parameters in MLP of Step 2 so that the total number of parameters is close to that of TAN.

### 4.2 PREDICTIVE PERFORMANCE (Q1)

The predictive accuracy on target test set is summarized in Table 3. For all the datasets, TAN achieves the best predictive performance outperforming the baseline methods. The accuracy boost of TAN compared to naive supervised learning (S(UL)) is up to 9.0%. This shows that using target data via transfer learning, even though they are unlabeled, helps improve the accuracy. Also, TAN shows much better accuracy than S(UL)-T(U) which further finetunes the model with the target unlabeled data. The reason is that S(UL)-T(U) causes the mismatch of distributions in the source and the target feature spaces. On the other hand, TAN uses the transfer alignment layer for matching the distributions which leads to superior performance in double blind domain adaptation.

### 4.3 EFFECT OF TRANSFER ALIGNER TRAINING (Q2)

We examine the effect of transfer aligner in TAN. Specifically, we verify the hypothesis that the superior performance of TAN does not come from its mere increase in layer size. To verify it, we train a competitor S(UL)-T(U)-Large which uses more layers compared to S(UL)-T(U) so that the same number of parameters is the same as that of TAN. Table 4 shows TAN consistently outperforms S(UL)-T(U)-Large with better accuracy up to 5.7%. This result verifies that the superior accuracy of TAN comes from its careful design of transfer aligner, not from its increase in the number of parameters.

### 4.4 EFFECT OF THE AMOUNT OF TARGET DATA (Q3)

We vary the ratio of target unlabeled data to total data by sampling the target unlabeled data and inspect the performance of TAN and competitors. For each dataset, we set the proportion of target

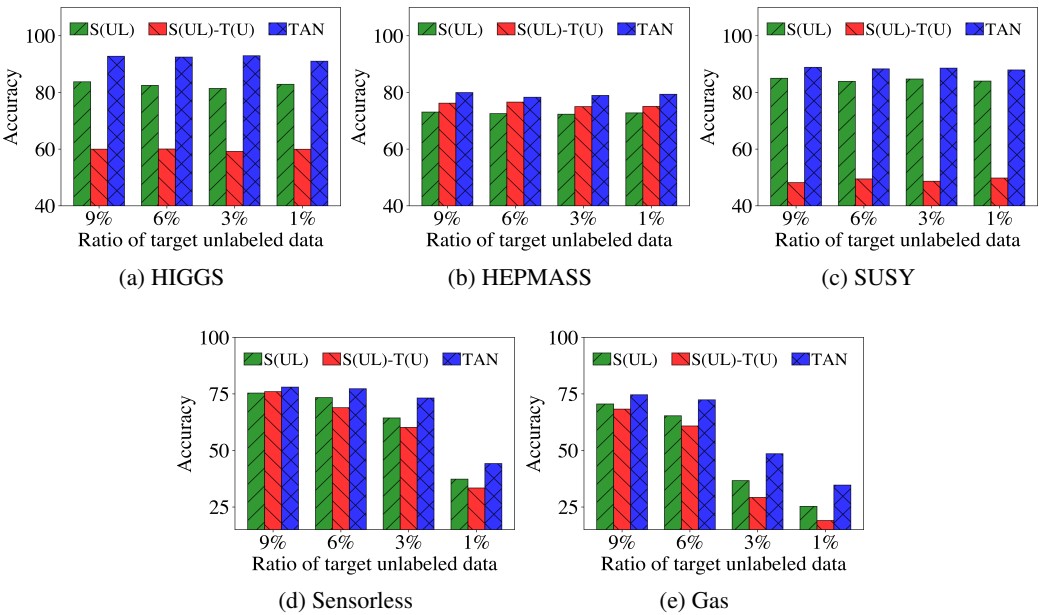

Figure 3: Test accuracy of TAN and baseline models for varying ratios of target unlabeled data. TAN gives the best accuracy regardless of the ratio of target unlabeled data in all datasets.

unlabeled data to 9%, 6%, 3%, 1% of the total data from both domains, to ensure that the numbers of target unlabeled instances are at least 100. As shown in Figure 3, TAN consistently outperforms baselines in all proportions and datasets, showing stability of transfer aligner in our method.

## 5  RELATED WORK

Training deep neural networks requires a massive amount of dataset; however, task-specific labeled dataset is often scarce. A common approach for the problem is to extract more information from other sources of data: unlabeled datasets from the same domain, or datasets from other domains.

Transfer learning (Pan & Yang (2010); Weiss et al. (2016)) aims to transfer a machine learning model in a source domain to a different target domain having different distribution from that of its training data. Generalizable knowledge learned from the source domain is leveraged to learn a better model in the target domain. There have been many researches applying transfer learning to achieve the state-of-the-art performance in both computer vision (Aytar & Zisserman (2011); Girshick et al. (2014); Long et al. (2015a); Ren et al. (2015)) and natural language processing (Collobert et al. (2011); Glorot et al. (2011); Chen et al. (2012); Xiao & Guo (2013)) domains.

Existing methods on deep transfer learning frequently minimize feature distribution discrepancy between source and target domains (Tzeng et al. (2014; 2015; 2017); Long et al. (2015b; 2016; 2017); Ganin & Lempitsky (2015); Ganin et al. (2016); Bousmalis et al. (2016); Zhang et al. (2015)). These methods utilize source and target data together in the training process to calculate and optimize feature distribution discrepancy. Other works on transfer learning iteratively assign and are trained on pseudo-labels for unlabeled source data and target data, respectively (Chen et al. (2011); Rohrbach et al. (2013); Lee (2013); Khamis & Lampert (2014); Sener et al. (2016); Ma et al. (2017); Saito et al. (2017); Cao et al. (2018)). These methods iteratively fine-tune models on both source and target data to provide better pseudo-labels. Most recently proposed works (Kumar et al. (2017); Dai et al. (2017); Miyato et al. (2017); Santos et al. (2017); Long et al. (2018)) exploit adversarial training to learn the structure in feature space. These studies measure the mismatch in data distribution by utilizing both source and target data. Unlike the aforementioned approaches, our goal is to perform domain adaptation where both of source and target data are not visible simultaneously.

Although existing blind domain adaptation method (Lampert (2014); Uzair & Mian (2017)) does not observe target data when using source data for training, they do not use the target data for training.

On the other hand, our goal is to solve double blind domain adaptation problem where both of source and target data are used for training, even though they are not visible simultaneously.

## 6 CONCLUSION

We propose TAN, a simple and novel deep transfer learning method for double blind domain adaptation. Since the target domain data have no labels, which makes it infeasible to train a classifier in the target domain, we utilize the classifier trained on a source domain to discriminate the target data. We minimize the domain discrepancy between source and target feature spaces, in order to make the classifier have better performance on the target domain data. However, source and target data cannot be visible simultaneously under the double blind constraint setting, which means general domain adaptation methods cannot be applied to the problem. To address this issue, TAN exploits transfer aligner that maps the target feature space to the source feature space without accessing any source data. Extensive experiments show that TAN provides the best transfer accuracy under double blind constraint. Future works include extending TAN to heterogeneous transfer learning.

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
