# OpenReview forum: "Transfer Alignment Network for Double Blind Unsupervised Domain Adaptation"
_ICLR.cc/2020/Conference — Reject_

### Official Review · AnonReviewer1 · 2019-10-12
**Official Blind Review #1**

**Rating:** 1

**Review:**



###Summary###

This paper tackles the transfer learning problem with the double-blind unsupervised domain adaptation, where either the source or the target domain cannot observe the data in the other domain, but data from both domains are used for training. The high-level intuition of this paper is based on the observation that in some practical settings, the transferring source data to the target domain is restricted due to the privacy policy. The goal is to learn a classifier which performs well in target classification task under double-blind constraint.

The setting of this paper is slightly different from the conventional domain adaptation. In this paper, the source domain has abundant unlabeled data and a small number of labeled data. The target domain only contains a limited number of unlabeled data.

The paper proposes a transfer alignment network (TAN) which comprises two autoencoders, one trained on the source domain and one trained on the target domain. In the domain adaptation phase, the model leverages an aligner to transfer the output of the target encoder to an aligned latent variable.  The aligner is trained to map the target code to source code on the target unlabeled data. The objective function is L2 distance between the source code and the mapped target code.

The whole pipeline is trained with four steps:
1) The source encoder and source decoder are trained with L2 reconstruction loss.
2) The source encoder and source classifier are trained with cross-entropy classification loss.
3) The target encoder and target decoder are trained with L2 reconstruction loss.
4) Train the aligner to map target code to source code on target unlabeled data with L2 distance loss.

The paper proposes to compare the TAN with three baselines: S(UL):  a stack of encoder and a neural network classifier trained using source data and tested on target data without finetuning. S(UL)-T(U): a model retrains the S(UL) with the target unlabeled data. S(UL)-T(U)-Large: a model which is similar to S(UL)-T(U) but contains more layers and parameters in MLP.

The experiments are performed on five multivariate datasets: HIGGS, HEPMASS, SUSY, Sensorless, and Gas.


### Novelty ###

The experimental setting proposed in this paper is interesting. However, the proposed model is trivial. The TAN model is composed of autoencoders and aligner. The training losses in the framework are L2 reconstruction loss and L2 distance loss. Thus, the novelty of this paper is incremental.

The experimental results in this paper are weak. First of all, the datasets used in this paper are not standard benchmarks. Secondly, the baselines in this paper are too trivial.


###Clarity###

Overall, the paper is well organized and logically clear. The images are well-presented and well-explained by the captions and the text.

###Pros###

1) The paper proposes an interesting transfer learning framework where either the source or the target domain cannot observe the data in the other domain.

2) The paper is applicable to many practical scenarios since the data privacy in the real-world application is critical.

3) The paper is overall well-organized. The claims of the paper are verified by the experimental results.

###Cons###

1) The paper proposes double-blind unsupervised domain adaptation as accessing the source and target domains is restricted in some practical settings. However, the source and target domain share the models trained on themselves, as well as the features extracted from the source domain and target domain data. The information about the original data can be recovered with the shared features and weights, which violates the settings proposed in this paper.

2) The main issue of this paper is the novelty is incremental. The proposed model is trivial as it only contains the auto-encoders and L2 loss.

3) The experimental part of this paper is weak. The datasets used in this paper are not the standard domain adaptation benchmark. It would be nice to see how does the proposed model work on standard domain adaptation benchmarks such as Office31, VisDA, Office-Home, DomainNet, etc.

VisDA: The Visual Domain Adaptation Challenge
https://arxiv.org/pdf/1710.06924.pdf
Office-Home : Deep Hashing Network for Unsupervised Domain Adaptation
http://hemanthdv.org/OfficeHome-Dataset/

The baselines used in this paper is also trivial. It is desirable to compare the proposed method with state-of-the-art domain adaptation methods.

Based on the summary, cons, and pros, the current rating I am giving now is "reject". I would like to discuss the final rating with other reviewers, ACs.



**Experience Assessment:**

I have published one or two papers in this area.

**Review Assessment: Checking Correctness Of Derivations And Theory:**

N/A

**Review Assessment: Checking Correctness Of Experiments:**

I carefully checked the experiments.

**Review Assessment: Thoroughness In Paper Reading:**

N/A

---

> ### Author Response · Authors · 2019-11-15
> **Response to reviewer 1**
>
> We thank the reviewer for the careful reading of the paper and their constructive comments. We would like to answer the reviewer’s questions as follows:
>
> 1. Problem setting
> Our goal is to improve the performance of the target task by transferring only the trained source model. Our definition of blind setting refers to not seeing the data. Of course, if we cannot see the model, there should not be anything to transfer. Transferring only model limits the leakage of privacy.
>
> 2. Novelty
> We proposed the unsupervised domain adaptation under the double blind setting, which is a challenging problem as it is difficult to train a target classifier properly only with the transferred source model. The setting is directly applicable to real-world settings due to privacy issues.
>
> 3. Datasets
> In this paper, we focused on experimenting with multivariate data. However, domain adaptation benchmarks have more complex data manifolds than multivariate data. It seems that a more complex architecture is needed to train domain adaptation benchmarks and we leave it as a future work.
>
> 4. Competitors
> We proposed the unsupervised domain adaptation under the double blind setting, where the source and target data cannot be visible in the training process simultaneously. Differently from our setting, the state-of-the-art unsupervised domain adaptation methods, e.g. MMD and DANN, train the model by feeding in the source and target data together. Thus, those existing methods cannot be used for baselines.

---

### Official Review · AnonReviewer3 · 2019-10-21
**Official Blind Review #3**

**Rating:** 1

**Review:**

This paper proposes a TAN model for double blind UDA problem, which supposes that partial data drawn from source domain is unlabeled and the target domain is completely unlabeled.

Actually, blind domain adaptation has been proposed several years ago. The double blind domain adaptation has no signficant difference.

From the model, I could not observe the technical novelty, although the authors focus on the "Aligner". Actually, autoencoder based domain adaptation has also been proposed for several years.

The aligner can actually be removed, by jointly training a domain-shared encoder in Step 3 when the unlabeled target data is used for training.
In other words, step 4 can be removed and in test stage, the output of encoder means the domain aligned feature representation.

Additionally, the source classifier is trained independently from the unlabeled target data. Although the domain aligned feature representation can be learned, it is still risky to directly apply the source classifier for unlabeled target data. Therefore, I suggest a safe strategy to train the source classifier and the domain aligned feature represenation module by jointly feeding the labeled source and unlabeled target data into the Step 2. That is, the step 3 can be integrated into step 2 for safer training.

I also have concerns on the experimental datasets. Why not use the benchmark visual datasets?

The proposed model lacks of comparisons with many state-of-the-art models.


**Experience Assessment:**

I have published in this field for several years.

**Review Assessment: Checking Correctness Of Derivations And Theory:**

I carefully checked the derivations and theory.

**Review Assessment: Checking Correctness Of Experiments:**

I carefully checked the experiments.

**Review Assessment: Thoroughness In Paper Reading:**

I read the paper thoroughly.

---

> ### Author Response · Authors · 2019-11-15
> **Response to reviewer 3**
>
> We thank the reviewer for the careful reading of the paper and their constructive comments. We would like to answer the reviewer’s questions as follows:
>
> 1. Novelty
> We proposed the unsupervised domain adaptation under the double blind setting, which is a challenging problem as it is difficult to train a target classifier properly only with the transferred source model. The setting is directly applicable to real-world settings due to privacy issues.
>
> 2. Architecture
> In this paper, we freeze the finetuned target encoder when training the aligner in Step 4. It seems feasible to directly get aligned features by jointly training the target encoder and the aligner, but it is not clear whether the joint training leads to good parameters. We will experiment with this architecture for a future work.
>
> 3. Training process
> We proposed the unsupervised domain adaptation under the double blind setting, where source and target data cannot be visible in the training process simultaneously. Therefore, we cannot train the source classifier by jointly feeding the labeled source and unlabeled target data in Step 2.
>
> 4. Datasets
> In this paper, we focused on experimenting with multivariate data. However, domain adaptation benchmarks have more complex data manifolds than multivariate data. It seems that a more complex architecture is needed to train domain adaptation benchmarks and we leave it as a future work.
>
> 5. Competitors
> We proposed the unsupervised domain adaptation under the double blind setting, where the source and target data cannot be visible in the training process simultaneously. Differently from our setting, the state-of-the-art unsupervised domain adaptation methods, e.g. MMD and DANN, train the model by feeding in the source and target data together. Thus, those existing methods cannot be used for baselines.

---

### Official Review · AnonReviewer2 · 2019-10-24
**Official Blind Review #2**

**Rating:** 1

**Review:**

In this paper, the authors claimed to address a new domain adaptation setting under double blind constraint, to meet the privacy requirement. Though the problem itself seems real and interesting, the solution in this work makes the problem quite trivial. In fact, any other domain adaptation method can be applied to address the problem, while the authors even did not compare with existing UDA methods which can be intuitively adapted.

Pros:
-	The problem that this work aims to address, i.e., domain adaptation under privacy constraint, seems reasonable and important.

Cons:
-	The authors proposed a quite trivial solution to solve this problem, which in turn makes all existing UDA adaptation methods off-the-shelf. For example, the most widely accepted DA algorithms, such as MMD and DANN, can be adapted to minimize the distance between the target encoder and the source encoder by training the weights for the target encoder. In this case, the aligner introducing more parameters is not even necessary.
-	The results are not promising and inspiring. S(UL), directly applying a model trained on a source dataset, has already achieved as competent results as TAN.
-	The datasets used are at a toy level from UCI. More profound discoveries are expected on DA benchmarks.
-	The paper needs significant proof-reading, as there are many grammatical errors and typos.


**Experience Assessment:**

I have published in this field for several years.

**Review Assessment: Checking Correctness Of Derivations And Theory:**

I carefully checked the derivations and theory.

**Review Assessment: Checking Correctness Of Experiments:**

I carefully checked the experiments.

**Review Assessment: Thoroughness In Paper Reading:**

I read the paper thoroughly.

---

> ### Author Response · Authors · 2019-11-15
> **Response to reviewer 2**
>
> We thank the reviewer for the careful reading of the paper and their constructive comments. We would like to answer the reviewer’s questions as follows:
>
> 1. Competitors
> We proposed the unsupervised domain adaptation under the double blind setting, where the source and target data cannot be visible in the training process simultaneously. Differently from our setting, the state-of-the-art unsupervised domain adaptation methods, e.g. MMD and DANN, train the model by feeding in the source and target data together. Thus, those existing methods cannot be used for baselines.
>
> 2. Results
> Table 3 compares the accuracy of S(UL) and TAN. The 2.64%~9% of improvements show the superiority of our method.
>
> 3. Datasets
> In this paper, we focused on experimenting with multivariate data. However, domain adaptation benchmarks have more complex data manifolds than multivariate data. It seems that a more complex architecture is needed to train domain adaptation benchmarks and we leave it as a future work.

---

### Decision · Program_Chairs · 2019-12-19

**Decision:**

Reject

**Comment:**

This paper tackles the problem of how to adapt a model from a source to a target domain when both data is not available simultaneously (even unlabeled) to a single learner. This is of relevance for certain privacy preserving applications where one setting would like to benefit from information learned in a related setting but due to various factors may not be willing to directly share data. The proposed solution is a transfer alignment network (TAN) which consists of two autoencoders (each trained independently on the source and the target) and an aligner which has the task of mapping the latent codes of one domain to the other.

All three reviewers expressed concerns for this submission. Of greatest concern was the experimental setting. The datasets chosen were non-standard and there was no prior work to compare against directly so the results presented are difficult to contextualize. The authors have responded to this concern by specifying the existing domain adaptation benchmarks are more challenging and require more complex architectures to handle the “more complex data manifolds”. The fact that existing benchmark datasets may be more complex the the dataset explored in this work is a concern. The authors should take care to clarify whether their proposed solution may only be applicable to specific types of data. In addition, the authors claim to address a new problem setting and therefore cannot compare directly to existing work. One suggestion is if using new data, report performance of existing work under the standard setting to give readers some grounding for the privacy preserving setting. Another option would be to provide scaffold results in the standard UDA setting but with frozen feature spaces. Another option would be to ablate the choice of L2 loss for learning the transformer and instead train using an adversarial loss, L1 loss etc. There are many ways the authors could both explore a new problem statement and provide convincing experimental evidence for their solution. The AC encourages the authors to revise their manuscript, paying special attention to clarity and experimental details in order to further justify their proposed work.